# *α*-Glucosidase Inhibitory and Antimicrobial Benzoylphloroglucinols from *Garcinia schomburgakiana* Fruits: In Vitro and In Silico Studies

**DOI:** 10.3390/molecules27082574

**Published:** 2022-04-15

**Authors:** Huy Truong Nguyen, Thanh-Trung Nguyen, Thuc-Huy Duong, Nguyen-Minh-An Tran, Chuong Hoang Nguyen, Thi-Hong-Anh Nguyen, Jirapast Sichaem

**Affiliations:** 1Faculty of Pharmacy, Ton Duc Thang University, Ho Chi Minh City 700000, Vietnam; nguyentruonghuy@tdtu.edu.vn; 2Institute of Research and Development, Duy Tan University, Da Nang 550000, Vietnam; trungnt@duytan.edu.vn; 3Faculty of Pharmacy, Duy Tan University, Da Nang 550000, Vietnam; 4Department of Chemistry, University of Education, 280 An Duong Vuong Street, District 5, Ho Chi Minh City 700000, Vietnam; huydt@hcmue.edu.vn; 5Faculty of Chemical Engineering, Industrial University of Ho Chi Minh City, Ho Chi Minh City 700000, Vietnam; 6University of Science, Vietnam National University Ho Chi Minh City, Ho Chi Minh City 700000, Vietnam; nhchuong@hcmus.edu.vn; 7Ho Chi Minh City University of Food Industry, 140 Le Trong Tan Street, Tay Thanh Ward, Tan Phu District, Ho Chi Minh City 700000, Vietnam; anhnth@hufi.edu.vn; 8Research Unit in Natural Products Chemistry and Bioactivities, Faculty of Science and Technology, Thammasat University Lampang Campus, Lampang 52190, Thailand

**Keywords:** *Garcinia schomburgakiana*, schomburgkianone I, benzoylphloroglucinol, *α*-glucosidase inhibition, antimicrobial inhibition, molecular docking model

## Abstract

α-Glucosidase plays a role in hydrolyzing complex carbohydrates into glucose, which is easily absorbed, causing postprandial hyperglycemia. Inhibition of α-glucosidase is therefore an ideal approach to preventing this condition. A novel polyprenylated benzoylphloroglucinol, which we named schomburgkianone I (**1**), was isolated from the fruit of *Garcinia schomburgkiana*, along with an already-reported compound, guttiferone K (**2**). The structures of the two compounds were determined using NMR and HRESIMS analysis, and comparisons were made with previous studies. Compounds **1** and **2** exhibited potent *α*-glucosidase inhibition (IC_50s_ of 21.2 and 34.8 µM, respectively), outperforming the acarbose positive control. Compound **1** produced wide zones of inhibition against *Staphylococcus aureus* and *Enterococcus faecium* (of 21 and 20 mm, respectively), compared with the 19 and 20 mm zones of compound **2**, at a concentration of 50 µg/mL. The MIC value of compound **1** against *S. aureus* was 13.32 µM. An in silico molecular docking model suggested that both compounds are potent inhibitors of enzyme α-glucosidase and are therefore leading candidates as therapies for diabetes mellitus.

## 1. Introduction

Diabetes mellitus (MD) is a set of metabolic conditions associated with excessive levels of blood glucose (hyperglycemia), which plays a pivotal function in the alleviation of long-term diabetic headaches [1]. Control of blood sugar levels is vital in diabetes therapy as it is associated with a marked decrease in headaches related to neuropathy, retinopathy, and cardiovascular conditions [2]. α-Glucosidase performs an essential function in carbohydrate digestion and glycoprotein biosynthesis. Inhibition of α-glucosidase involves certain small intestinal membrane enzymes: maltase-glucoamylase (MGAM) and sucrase-isomaltase (SI); these are implicated in the breakdown of nutritional sugars and starches. Inhibition of MGAM and SI helps manage blood glucose levels in patients with type 2 diabetes [3,4]. Acarbose, miglitol, and voglibose are the currently preferred α-glucosidase inhibitors for the control of diabetes [5]. However, their long-term consumption can increase unwanted side effects, which include stomach pain, bloating, diarrhea, and flatulence [6]. To avoid such unfavorable effects and enlarge the arsenal of therapies, novel α-glucosidase inhibitors are always being sought. Many inhibitors, including flavonoids, terpenoids, alkaloids, and phenolic compounds, have been isolated from medicinal plants [7]. Antibiotics have saved millions of lives each year since the discovery of penicillin. However, many organisms have evolved resistance to widely used antibiotics, such as methicillin-resistant *Staphylococcus aureus* and multidrug-resistant and extensively drug-resistant *Mycobacterium tuberculosis*. Replacement drugs with effectiveness against multidrug-resistant and extensively drug-resistant strains are desperately needed. Natural products are promising starting points for clinical antimicrobials and may be isolated from microorganisms or plants [8].

The identification of novel chemical entities with antibacterial properties must be conducted alongside the implementation of policies to restrict the inappropriate and illogical use of antibiotics. Docking methods can be used to predict optimal ligand/receptor conformations. Macromolecular analysis applies scoring functions and changes in free energy upon binding [9]. The method is validated from the RMSD value [10]. In silico molecular docking models have been applied to ligand processes such as α-glucosidase enzyme inhibition [11,12,13,14], antimicrobial activity [15,16,17], anticancer activity [18], antioxidant activity [19], anti-inflammatory activity [20,21], and acetylcholinesterase inhibition.

*Garcinia*, a member of the Clusiaceae family, is a sturdy evergreen tree and shrub common in moist, tropical lowland forests in Asia, Africa, South America, Australia, and Polynesia. Phytochemically, this genus is mainly known for xanthones, bioflavonoids, triterpenoids, and prenylated phloroglucinols, which exhibit a wide range of bioactivities. In particular, prenylated phloroglucinols have attracted considerable attention due to their wide structural diversity and interesting biological activities. *Garcinia schomburgakiana* Pierre, known as Ma-dan in Thai, is an edible evergreen tree that grows in Laos, Vietnam, Cambodia, and Thailand [21]. In ethnomedicine, it is used as a laxative and expectorant and in the treatment of coughs, menstrual disorders, and diabetes [22]. Previous studies on the bioactive constituents of *G. schomburgkiana* have reported the presence of flavonoids, xanthones, triterpenoids, depsidones, phloroglucinols, and biphenyl derivatives. Some of these secondary metabolites showed antimalarial, cytotoxic, and anti-α-glucosidase properties [21,23]. However, there have been few phytochemical studies of *G. schomburgkiana* fruit. Le and co-workers reported the first isolation of polyprenylated benzoylphloroglucinols and biflavonoids from the fruit of this plant [24]. The discovery of these beneficial secondary metabolites encouraged us to investigate the fruit of *G. schomburgkiana* more extensively. Herein, we describe the isolation and the characterization of a novel polyprenylated benzoylphloroglucinol, schomburgkianone I (**1**), along with a known compound, guttiferone K (**2**) [24]. Both compounds were evaluated for α-glucosidase inhibitory and antimicrobial activities. An in silico molecular docking model was constructed to investigate the mechanisms of inhibition.

## 2. Results and Discussion

### 2.1. Phytochemical Identification

Compound **1** was obtained as a yellow gum, and its molecular formula was determined to be C_38_H_50_O_6_ by HRESIMS and from the NMR spectra (Table 1, the spectrum can be seen in Appendix A). The ^1^H NMR data showed signals of a 1,2,4-trisubstituted benzene ring (δ_H_ 7.38 (1H, d, *J* = 2.0 Hz), 7.19 (1H, dd, *J* = 8.5, 2.0 Hz), and 6.85 (1H, d, *J* = 8.5 Hz)), three olefinic protons (δ_H_ 4.84, 5.04, and 5.21), and six downfield methyls (δ_H_ 1.75, 1.66, 1.64, 1.63, and 1.59 × 2) characteristic of three isoprenyl groups, three upfield singlet methyls (δ_H_ 1.25, 1.01, and 0.82), and other methylene and methine protons in the range 1.43–2.66 ppm. The ^13^C NMR in accordance with the HSQC spectrum indicated the presence of 38 carbon signals, including three ketone carbons (δ_C_ 208.4, 191.7, and 189.8), three aromatic methines (δ_C_ 125.6, 116.1, and 115.5), three olefinic methines (δ_C_ 125.6, 123.7, and 122.2), and eight sp^2^ quaternary carbons (δ_C_ 179.0, 150.6, 146.8, 133.8, 133.1, 131.7, 131.3, and 116.6, the first three being oxygenated). The full analysis of both 1D (^1^H and ^13^C) and 2D (COSY, HSQC, and HMBC) NMR spectra suggested that compound **1** contained a benzoylphloroglucinol skeleton (Figure 1), and its NMR spectra were also close to those of guttiferone K (**2**), a major constituent of this plant material [24], except for the change of the substituent at C-8. The isoprenyl group at C-8 in compound **2** was replaced by an isopentyl group in compound **1**. This was further confirmed by the two methyls at C-32 and C-33 being upfield-shifted (δ_H_ 1.25 and 1.01). HMBC correlations of these methyls to the oxygenated carbon at δ_C_ 82.0 indicated the attachment of an oxygen atom at C-31. The etherification between C-31 and C-1 was defined based on an analysis of ^13^C NMR data. The absence of the hydrogen-bond hydroxyl group at C-1 would affect the carbons at C-1, C-3, and C-10. In particular, the chemical shifts of these carbons were upfield-shifted compared to those of compound **2** (C-1/C-3/C-10: δ_C_ 179.0/189.8/191.7 in compound **1** vs. δ_C_ 199.3/198.6/195.6 in compound **2**). This chemical feature was similar to previously reported benzoylphloroglucinols isogarcinol [25], garcinialiptone B [26], and cycloxanthochymol [26]. HMBC and ^1^H-^1^H COSY correlations (Figure 2) provided complete NMR assignments of compound **1**.

The relative configuration of compound **1** was determined using NOESY correlations. The same orientation of the isoprenyl moiety at C-4 and the isopentyl group at C-8 was determined by NOESY correlations of all H_2_-17 and H_3_-21 to the same proton H-15. NOESY correlations between H_2_-17 and H_2_-23, as well as H_2_-23 and H_2_-24, revealed that the CH_2_-17, CH_2_-23, and CH_2_-24 side chains were all oriented in the same direction. This implied that the methyl group (H_3_-22) was at the opposite site. The stereochemistry of compound **1** was similar to that of co-isolate **2** and other benzoylphloroglucinol derivatives from the same species [24]. Therefore, the chemical structure of compound **1** was concluded to be a new polyprenylated benzoylphloroglucinol, which we named shomburgkianone I.

### 2.2. Biological Activities of Isolated Compounds

The in vitro *α*-glucosidase inhibitory activity of compounds **1** and **2** was evaluated. Compounds **1** and **2** displayed significant *α*-glucosidase inhibitory activity with IC_50_ values of 21.2 and 34.8 µM, respectively, which were superior to that of the positive drug acarbose (IC_50_ 332 µM). The presence of the pyran ring at C-1 and C-8 in the case of compound **1** might be responsible for enhancing the activity.

Compounds **1** and **2** were evaluated for their antimicrobial activity against antibiotic-resistant, pathogenic bacteria *S. aureus*, *E. faecium*, and *A. baumannii*. Both compounds **1** and **2** inhibited *S.*
*aureus* with inhibition zones of 21 and 19 mm, respectively, at a concentration of 50 µg/mL. They also inhibited *E. faecium* with inhibition zones of 20 mm for both compounds at the tested concentration but failed to show any activity against *A*. *baumannii*. The MIC value of compound **1** against *S.*
*aureus* was 13.32 µM, compared to the positive control, kanamycin (MIC 8.26 µM). In addition, compound **1** exhibited weak cytotoxicity toward the HEK293 normal cell line with an IC_50_ value of 87 µM.

Benzoylphloroglucinols derived from *Garcinia* species exhibited good cytotoxicity against many cancer cell lines [26,27]. For example, compounds from *Garcinia multiflora* had apoptosis-inducing effects against HeLa-C3 cells, and also had strong HeLa cell growth inhibition effects with IC_50_ values in the range of 12.4–23.0 µM. Several investigations regarding the antimicrobial activity of benzoylphloroglucinols were reported [28,29]. Guttiferone BL, a derivative of compound **2**, showed low activity against *S.*
*aureus*, indicating the important role of the number of isoprenyl units in the activity [29].

### 2.3. Molecular Docking Studies of Compounds **1** and **2**

#### 2.3.1. In Silico α-Glucosidase Enzyme Inhibition

Ranked pose 148 (Compound **1**): Pose 148 is the most stable conformation of compound **1** and was selected from among 200 poses or models to build a simulation of the thermodynamic site and ligand interactions. This pose docked to enzyme 4J5T (PDB) at the active site of the enzyme with the values of affinity energy, ΔG^o^, and inhibition constant, Ki, of −10.51 Kcal·mol^−^^1^ and 0.02 µM, respectively, as shown in Table 2. Pose 148 formed two hydrogen bonds from active atoms on pose 148 to Glu 429 and Arg 428 on the enzyme, as seen in Figure 3 and Table 2. At the thermodynamic site, pose 148 was the best docking pose among the considered poses for compounds **1**, **2**, and acarbose. The significant ligand interactions between pose 148 and 4J5T are depicted in Figure 3, Figure 4 and Figure 5 and Table 2. Pose 148 interacted well with enzyme 4J5T because it was identified as full of three parts of ligand interactions, namely the capping unit, connecting unit (CU), and functional group [30], as seen in Figure 4. The capping unit of pose 148 was identified as a protein via one pi-cation from Arg428 to the pi-electron system of the aromatic ring, an alkyl, or pi-alkyl from Leu 563 to the pi-electron system of the aromatic ring, and pi–pi stacking from Phe 444 to the pi-electron system of the aromatic ring. The connecting unit of pose 148 was detected by one pi-sigma from Tyr 709 to the allyl group in pose 148; an alkyl or pi-alkyl from Trp 715, Trp 789, Trp 710, and Trp 391 to the alkenyl group in pose 148; an alkyl or pi-alkyl interaction from Trp 391, Phe 389, and Arg 428 to allyl groups in this pose; and one pi-sigma interaction from Phe 385 to the methylene group of the oxygen heteroatom ring in pose 148. The functional group consisted of one hydrogen bond from Phe 444 to the hydrogen atom of the phenolic hydroxyl ring. Pose 148 is considered the best docking pose because of its thermodynamic site and full ligand interactions. Ranked poses determined pose 148 (compound 1) > pose 41 (compound 2) > acarbose (standard drug). drug). Regarding other secondary interactions, one ligand map indicated the strength of ligand interaction between the best-ranked pose, pose 148, and enzyme 4J5T during the processing of the pose and receptor 4J5T. We included interactions such as hydrogen bonds and steric, electrostatic, and overlap interactions, as shown in Figure 5. As seen in Figure 5, there are many steric interactions and hydrogen bonds formed between pose 148 and the 4J5T target enzyme. This proved that pose 148 and 4J5T interacted strongly due to more residual amino acids forming around pose 148. As seen in Figure 5, the hydrogen bonds are depicted by brown dashed lines from the residual active amino acid Glu 429 to oxygen atoms of the phenolic hydroxyl groups of the phenyl ring. The steric interactions (green dashed lines) linked residual active amino acids such as Phe 385, Phe 389, Arg 428, Glu 429, Phe 444, Asn 453, Glu 566, Asp 568, Glu 771, and Trp 789 to active atoms in pose 148. The overlap interactions are demonstrated by the violet circles. The stronger the overlap interactions between active atoms on the ligand and enzyme 4J5T, the bigger diameter of the violet circle.

The results of the molecular docking model indicated that the pharmacophore of pose 148 or compound **1** was relative to the phenyl ring, p-hydroxyl phenyl ring, methyl group (C-20), methylene group (C-31), two methyl groups (C-28 and C-29), and 3-methylbut-2-en-1-yl group (C-34–38), as shown in Figure 6. Ranked pose 41: Pose 41 is the most stable conformation of compound **2** and was selected from among 200 poses or models to build a simulation of the thermodynamic site and ligand interactions. It interacted with the active site of enzyme 4J5T with the thermodynamic parameters affinity energy, ΔG^o^, and inhibition constant, K_i_, of −10.12 Kcal·mol^−^^1^ and 0.04 µM, respectively as seen in Table 2. The results of fundamental ligand interactions in the interaction model of pose 41 and the 4J5T enzyme are presented in Table 2 and Figure 7, Figure 8 and Figure 9. This pose bound three hydrogen bonds from active residual amino acids Arg428 and Glu429 to active atoms in the pose, as shown in Table 2 and Figure 7. The significant ligand interactions between pose 41 and 4J5T are indicated in Figure 8, and this pose identified good ligand interactions because three parts of the ligand (capping group, connecting unit, and functional group) have fully characteristic ligand interactions. The capping group of poses is revealed by one pi-alkyl from Phe 444 to the pi-electron system of the phenyl ring. The connecting unit or linker of the pose is revealed by pi-alkyls from His 561 and Tyr 709 to the allyl group, pi-alkyls from Trp 391 and Phe 389 to the carbon atom of the alkenyl group, pi-alkyls from Trp710 and Trp715 to the pi-electron system of the alkenyl group, and pi-sigma from Phe 385 and Phe 389 to the carbon atom of the methyl group. The functional group of this pose is revealed by hydrogen bonds from Glu 429 and Phe 444 to hydrogen atoms of the phenolic hydroxyl group of the benzene ring. Pose 41 was considered to have good ligand interactions with target enzyme 4J5T, but it has weaker ligand interactions than pose 148 or compound **1**, due to the thermodynamic site (higher affinity energy values, ΔG^o^). As shown in Figure 9, one ligand map indicated more steric interactions forming between pose 41 and the enzyme. It proved that ligand interactions between ranked pose 41 and enzyme 4J5T are very strong. The residual amino acids are relative to steric interactions such as Phe 389, Trp391, Asp 392, Arg 428, Glu 429, Phe 444, His 561, Asp 568, Asp 569, Tyr 709, Trp 710, Trp 715, Glu 771, and Trp 789. There is one hydrogen bond from amino acid Glu 429 to the oxygen atom of the phenolic hydroxyl of the aromatic ring in pose 41.

The overlap interactions are represented by violet circles. The size of the violet circles has increased, as have the overlap interactions. As shown in Figure 10, the pharmacophore of pose 41 or compound **2** is determined as one phenolic ring, one alkenyl group, two allyl groups, one alkenyl group, and one vinyl carbon atom. Ranked pose 170: The results of docking calculations are presented in Table 2 and Figure 11, Figure 12 and Figure 13. As shown in Table 2, pose 170, the most stable conformation ligand of acarbose, docked to the active center of the enzyme with the values of affinity energy, ΔG^o^, and inhibition constant, K_i_, of −5.22 Kcal·mol^−^^1^ and 149.6 µM, respectively. There are 10 hydrogen bonds that formed from residual amino acids to active atoms in pose 170, as seen in Table 2 and Figure 11. As shown in Figure 12, the functional groups of this pose are hydrogen bonding linked from Tyr 709, Gly 566, Glu 771, Asp 392, and Trp 710 to the hydrogen atom of the hydroxyl group in this pose. The connecting unit and capping group have no ligand interactions from amino acids to the pose. At the thermodynamic site, pose 170 was bound weakly with the enzyme due to the value of affinity energy, ΔG^o^. The ligand map showed the secondary interactions between pose 170 and enzyme 4J5T, as shown in Figure 13. Those interactions were hydrogen bonds (brown lines) from Asp 392, Glu 771, Asp 568, Tyr 710, Tyr 709, and Gly 566 to active atoms in pose 170, steric interactions (green lines), and overlap interactions (violet circles). More amino acids are built around this pose. It proved that ligand interactions between this pose and the enzyme are strong.

The in silico docking enzyme glucosidase inhibition model was validated as follows: As shown in Figure 14 and Figure 15, poses 148 (green), 41 (violet), and 170 (yellow) are docked to the same active center on enzyme 4J5T. As shown in Table 3, poses 148 and 41 aligned to reference pose 170 (ligand acarbose). The pose-pair RMSD values for poses 148 and 170 and poses 141 and 170 were calculated as 2.280 and 4.094 Å, respectively, by PyMOL software. These values demonstrated the molecular docking model’s validation in redocking, docking orientation, conformation, and interactions [30]. These conditions give rise to predictions about ligand interactions with a compound of interest.

#### 2.3.2. In Silico Antimicrobial Activity

Pose 158, one of 200 ligand conformations immersed in receptor 2VF5, was one of the ranked poses. One enzyme, glucosamine-6-phosphate synthase, synthesizes glucosamine-6-phosphate. It is a good target in antimicrobial chemotherapy. This enzyme participates in the biosynthesis of an amino sugar, namely uridine 5′-diphospho-N-acetyl-D-glucosamine (UDP-GlcNAc). UDP-GlcNAc was discovered in bacterial and fungal cell walls. Inactivation of GlcN-6-P synthase for a short period is very dangerous for fungal cells [18]. All significant calculations of ligand interactions between this pose and 2VF5 are presented in Table 4 and Figure 16, Figure 17 and Figure 18. As seen in Table 4, pose 158 is anchored to 2VF5 with the values of affinity energy, ΔG^o^, and inhibition constant, K_i_, of −8.56 Kcal·mol^−^^1^ and 0.53 µM. Pose 158 formed three hydrogen bonds from Ala 496 to active atoms in pose 158, as seen in Table 4 and Figure 16. As shown in Figure 17, the fundamental ligand interactions between pose 158 and target enzyme 2VF5 are indicated in one 2D diagram. Pose 158 interacted well with enzyme 2VF5 because three parts of this pose interacted well with enzyme 2VF5. The capping unit of this pose is revealed by one alkyl or pi-alkyl from Leu 480 to the pi-electron system of the phenyl ring. A connecting unit or linker was detected via pi-alkyl or alkyl from Leu 484, Cys 300, and Ile 326 to the alkenyl group and the methyl on pose 158. The functional group of this pose formed hydrogen bonds from Ala 496 to hydrogen atoms of the phenolic hydroxyl group of the phenyl ring. As seen in Figure 18, the ligand map indicated the secondary interactions such as hydrogen bonds (Ala 496, brown lines), steric interactions (Glu 495, Lys 487, Leu 480, Gly 301, and Asn 305, green lines), and overlap interactions (violet circles). Pose 35 was ranked among 200 conformations linked to enzyme 2VF5 with the values of −6.24 Kcal·mol^−^^1^ and 26.84 µM, as shown in Table 4 and Figure 19. This pose formed three hydrogen bonds from Val 324 and Tyr 304 to active atoms on enzyme 2VF5, as seen in Table 4 and Figure 19. As shown in Figure 20, the significant ligand interactions are presented in one 2D diagram between pose 35 and 2VF5.

Pose 35 cannot interact well with an enzyme because the capping group of this pose (aromatic ring) has no ligand interactions. As seen in Figure 21, one ligand map indicated the secondary interactions such as hydrogen bonding (Tyr 304 and Val 324, brown lines) and steric interactions (Leu 480, Tyr 476, Tyr 304, Val 324, and Lys 487). Other interactions are overlap interactions (violet circles). The steric interactions formed around this pose demonstrated that pose 35 formed a strong bond with the enzyme. Pose 172 is the ranked pose of ligand apramycin, a standard drug, docked to the active site of enzyme 2VF5 with the values of affinity energy, ΔG^o^, and inhibition constant, K_i_, of −6.94 Kcal·mol^−^^1^ and 8.17 µM, respectively, as shown in Table 4. All significant interactions between this pose and 2VF5 are exposed in Figure 22, Figure 23 and Figure 24. Pose 172 formed 10 hydrogen bonds from Ser 316, Ala 520, Asp 474, Glu 569, and Tyr 312 to active atoms in pose 172, as seen in Table 4 and Figure 22. As shown in Figure 23, the important ligand interactions between pose 172 and enzyme 2VF5 are indicated in one 2D diagram, as seen in Figure 23. Due to the short interactions between the connecting unit and the capping group, the pose does not interact well with the enzyme. They are only electrostatic and hydrophilic interactions. As shown in Figure 24, the secondary interactions between pose 172 and 2VF5 are strong due to more steric interactions that build around the ligand. Pose 83 is the ranked pose of one small ligand, which is available in enzyme 2VF5. All fundamental ligand interactions are included in Table 4 and Figure 25, Figure 26 and Figure 27. Pose 83 docked to the enzyme at the active site with the values of affinity energy, ΔG^o^, and inhibition constant, K_i_, of −5.38 and 114 µM, respectively.

As seen in Table 4 and Figure 25, pose 83 formed 13 hydrogen bonds from residual amino acids such as Thr302, Gln 348, Ser 349, Thr 352, Ser 401, Glu 488, Ala 602, Ser 349, and Lys 603 to active atoms on pose 83. As shown in Figure 26, the significant interactions between pose 83 and 2VF5 are exposed in one 2D diagram, and this pose cannot interact well with the enzyme 2VF5 because the capping group and connecting unit had no ligand interactions with the enzyme. As seen in Figure 27, the ligand map showed hydrogen bonds (Lys 603 and Ser 604, brown lines) and steric overlaps (Ser 401, Gln 348, Ser 349, Thr 352, Lys 603, Ala 602, and Thr 302, green lines). The steric interactions indicated weak interactions between pose 83 and 2VF5. The silico docking model for antimicrobial activity was validated as follows: As shown in Table 5, the pose-pair RMSD values were calculated as 3.320, 2.166, and 2.839 for poses 158 and 172, poses 35 and 172, and poses 83 and 172, respectively. As shown in Figure 28, poses 158 (red), 35 (cyan), and 83 (violet) are aligned to pose 172 (yellow, a standard drug). As indicated in Figure 29, ranked poses 158 (red), 35 (cyan), 83 (violet), and 172 (yellow) are docked to the same active site on enzyme 2VF5. The values of RMSD between pose pairs and docking of each pose to the same active site of an enzyme proved the validation of the molecular docking model at sites such as binding sites, orientation, conformation, and bonding. For the in silico docking model of antimicrobial activity, pose 158 (compound 1) is the best docking pose among poses 35, 172, and 83 in thermodynamic and molecular docking.

#### 2.3.3. In Silico Physicochemical Properties, Drug-Likeness, and Pharmacokinetic Predictions

In silico physicochemical properties, drug-likeness, and pharmacokinetic predictions are indicated in Table 6, Table 7, Table 8, Table 9, Table 10, Table 11, Table 12, Table 13 and Table 14. As shown in Table 6, the physicochemical properties such as molecular weight, Van der Waals volume, the number of hydrogen bond acceptors, the number of hydrogen bond donors, the number of rotatable bonds, the number of atoms in the biggest ring, the number of heteroatoms, formal charge, flexibility, stereocenters, topological polar surface area, logS, logP, and logD are in ranges. As seen in Table 7, most of the parameters of medicinal chemistry are in scope except for QED, SAscore, Lipinski rule, and GSK rule. As indicated in Table 8, compound **1** has good absorption according to parameters of Caco-2 permeability, MDCK, Pgp-inhibitor, Pgp-substrate, and HIA. The drug distribution of compound **1** was determined well by plasma protein binding, volume distribution, and blood–brain barrier penetration variables in Table 9. As seen in Table 10, the properties of drug metabolism of compound **1** were detected as being in permissible ranges. The parameters of the drug excretion such as CL and T1/2 of compound **1** are reported in ranges as shown in Table 11. As indicated in Table 12, the results for the drug toxicity of compound **1** indicated that three variables, namely human hepatotoxicity, drug-induced liver injury, and respiratory toxicity are out of scope. The properties of the environmental toxicity of compound **1** presented in Table 13 are in the expected ranges. Toxicophore rules proved that the parameters are in expected ranges, as presented in Table 14. The predictions of physicochemical properties, drug-likeness, and pharmacokinetics indicate that compound **1** has potential drug-likeness in in silico docking.

## 3. Materials and Methods

### 3.1. Source of the Plant Material

Fruits of *G. schomburgkiana* were collected in October 2021 in Mueang Nakhon Nayok District, Nakhon Nayok Province, Thailand. The identification was confirmed by Dr. Suttira Sedlak, Walai Rukhavej, Botanical Research Institute, Mahasarakham University, Thailand. A voucher specimen Khumkratok No. 92-08 has been deposited in the Walai Rukhavej Botanical Research Institute, Mahasarakham University, Thailand.

### 3.2. Isolation

Dried fruits of *G. schomburgkiana* (5 kg) were exhaustively extracted with acetone (10 L × 3, 12 h) at ambient temperature. After evaporation, the crude extract (501 g) was partitioned with n-hexane and n-hexane–EtOAc (1:1, *v*/*v*), yielding n-hexane (23 g, H) and n-hexane–EtOAc (1:1) (47 g, HEA) extracts, respectively. The water-containing solution was evaporated to afford the extract A (394 g). The H extract (23 g) was subjected to silica gel column chromatography (CC) with n-hexane–EtOAc–acetone (4:1:0.2, *v*/*v*/*v*) as an eluent to provide fractions H1–H4. Fraction H4 (3.8 g) was purified using silica gel column chromatography (CC) and eluted with n-hexane–chloroform–acetone (7:4:6, *v*/*v*/*v*), affording fractions H4.1–H4.5. Fraction H4.2 (301 mg) was subjected to C18 reverse-phase silica gel CC with a solvent system of MeOH–H_2_O (15:1, *v*/*v*), yielding subfractions H4.2.1–H4.2.3. Fraction H4.2.1 was rechromatographed using C18 reverse-phase silica gel CC with the same chromatographic method to afford compound **1** (3.1 mg). Fraction H4.4 (467 mg) was subjected to C18 reverse–phase silica gel CC and eluted with MeOH–H_2_O (20:1, *v*/*v*) to give compound **2** (48 mg).

**Schomburgkianone I (1).** Colorless gum. [α]^20^_D_ +115 (c 0.1, CHCl_3_). HRESIMS m/z 601.3534 [M−H]^−^ (calcd. for C_38_H_49_O_6_ 601.3529); ^1^H NMR (acetone-*d*_6_, 500 MHz) and ^13^C NMR (acetone-*d*_6_, 125 MHz) see Table 1.

### 3.3. α-Glucosidase Inhibition Assay

The α-glucosidase (0.2 U/mL) and substrate (5.0 mM p-nitrophenyl-α-D-glucopyranoside) were dissolved in 100 mM pH 6.9 sodium phosphate buffer [31]. The inhibitor (50 µL) was preincubated with α-glucosidase at 37 °C for 20 min, and then the substrate (40 µL) was added to the reaction mixture. The enzymatic reaction was carried out at 37 °C for 20 min and stopped by adding 0.2 M Na_2_CO_3_ (130 μL). Enzymatic activity was quantified by measuring absorbance at 405 nm. All samples were analyzed in triplicate at five different concentrations around the IC_50_ values, and the mean values were retained. The inhibition percentage (%) was calculated by the following equation: Inhibition (%) = (1 − (A_sample_/A_control_)) × 100.

### 3.4. Antibacterial Activity Assay

The agar well diffusion method was used to evaluate the antibacterial activity of the isolated compounds on antibiotic-resistant, pathogenic bacteria *Staphylococcus aureus*, *Enterococcus faecium*, and *Acinetobacter baumannii*. Three bacterial pathogens were cultured in nutrient broth at 37 °C for 18 h. The cultures were diluted with sterile 0.9% NaCl to obtain bacterial solutions of 1.5 × 10^8^ CFU/mL. This solution with a volume of 100 μL was spread on a Mueller–Hinton agar plate. Holes with a diameter of 8 mm were punched aseptically to create wells on the surface of the Mueller–Hinton agar. The compounds were dissolved in DMSO. The amount of 50 µg of each compound solution was inserted into the wells. The plates were incubated at 37 °C for 16–18 h, and the antibacterial activity of each compound was recorded by measuring the diameters of the inhibition zones surrounding the wells. DMSO was used as a control [32]. MIC values were recorded as the lowest concentrations of compounds **1** and **2** that inhibited the growth of *S. aureus*. Kanamycin was used as the positive control in this experiment.

### 3.5. Cytotoxicity Assay

The cytotoxic evaluation of compound **1** against the HEK293 normal cell line was applied from a previous procedure [23].

### 3.6. Molecular Docking Studies and ADMET

In silico molecular docking models for α-glucosidase enzyme inhibition and antimicrobial activity were performed as shown in Appendix A. The α-glucosidase enzyme inhibition was conducted between ligands and receptor 4J5T (PDB) [30]. The file of grid parameters for α-glucosidase enzyme inhibition was set up by spacing, elements, and activity center, which were 0.5 Å, 60 × 60 × 60, and (X, Y, Z = −18.418, −20.917, 8.049). For antimicrobial activity, one receptor was used: 2VF5 (PDB) [18]. The grid parameter file for antimicrobial activity was determined by spacing, elements, and activity center, which were 0.5 Å, 60 × 60 × 60, and (X, Y, Z = 26.579, 22.731, 8.113). The docking parameter input and output files were Generic Algorithm and Lamarckian parameters. The values of the RMSD of models were calculated by PyMOL software. AMDME was used to predict the drug properties of compound **1** such as absorption, distribution, metabolism, excretion, and toxicity based on the new online version of ADMET, ADMETlab 2.0 (ADMETlab 2.0 (scbdd.com), accessed on 2 April 2022). The profile of one drug, compound **1**, was evaluated in the following article: Pharmacokinetics and drug-likeness of antidiabetic flavonoids: Molecular docking and DFT study (plos.org) (accessed on 2 April 2022).

## 4. Conclusions

This is the first report on α-glucosidase inhibitory and antimicrobial activities of isolated benzoylphloroglucinols present in the fruits of *G. schomburgakiana*. Compounds **1** and **2** showed powerful yeast α-glucosidase inhibitory activity, which was superior to that of a positive agent. On the other hand, compound **1** had the maximum zone of inhibition against *S. aureus* and *E. faecium* (21 and 20 mm, respectively), whereas compound **2** showed a maximum zone of inhibition toward both bacteria (19 and 20 mm, respectively) at the concentration of 50 µg/mL. Both in vitro and in silico study results suggest the potential of *G. schomburgakiana* fruits for future application in the treatment of diabetes, and active compounds **1** and **2** have emerged as promising molecules for therapy.

## Figures and Tables

**Figure 1 molecules-27-02574-f001:**
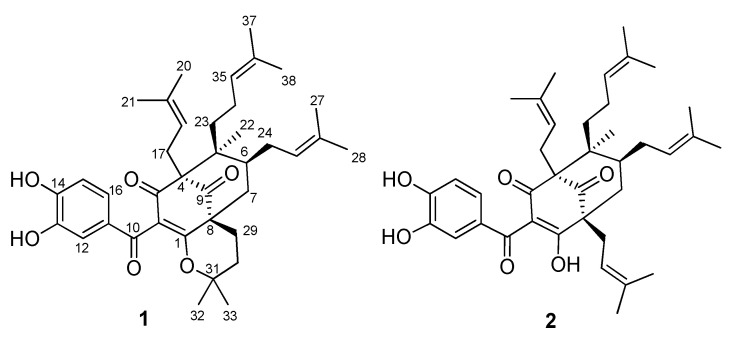
Chemical structures of compounds **1** and **2**.

**Figure 2 molecules-27-02574-f002:**
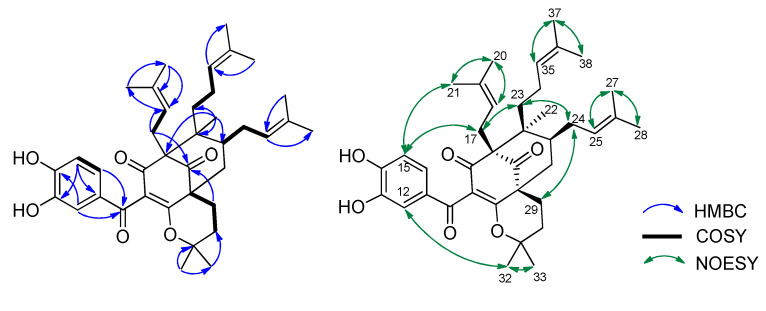
Key COSY, HMBC, and NOESY correlations of compound **1**.

**Figure 3 molecules-27-02574-f003:**
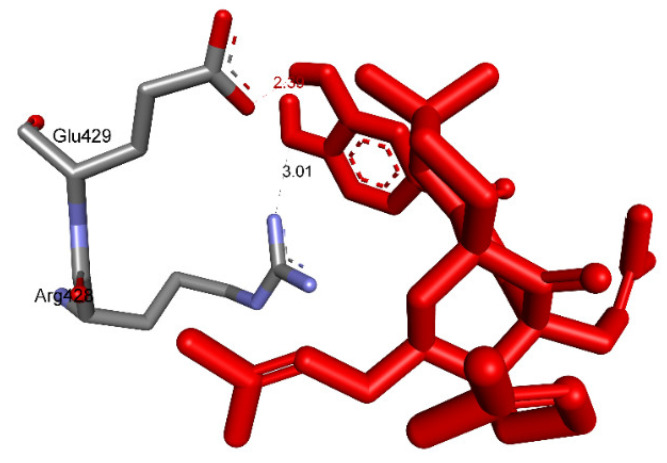
The hydrogen bonding formed from atoms on active pose 148 to atoms active on residual amino acids.

**Figure 4 molecules-27-02574-f004:**
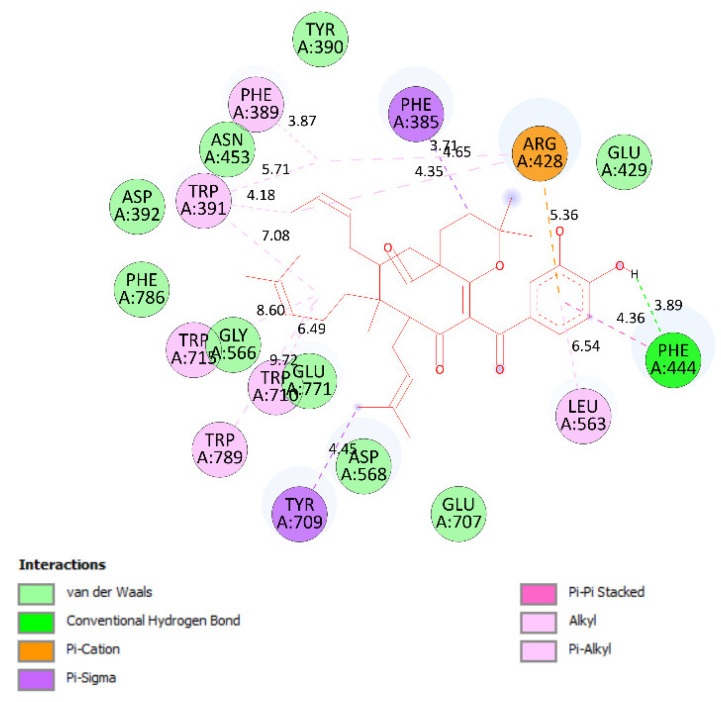
The most significant ligand interactions between compound **1,** pose 148 and receptor 4J5T (PDB), a member of the α-glucosidase enzyme family.

**Figure 5 molecules-27-02574-f005:**
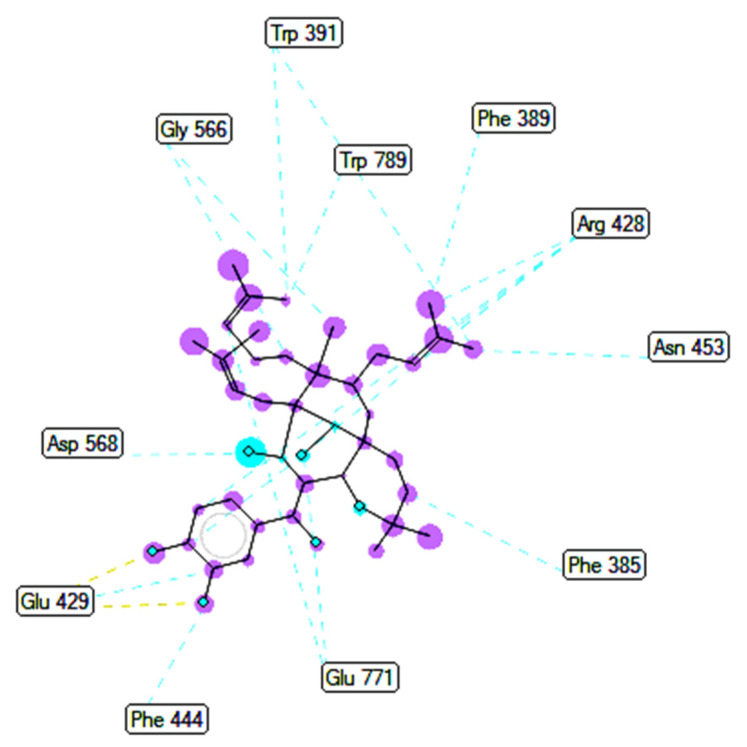
The ligand map shows the secondary interactions such as hydrogen bonds (yellow), steric interactions (light green), and overlap interactions (circle violet) between residual amino acids on enzyme 4J5T and pose 148, compound **1**.

**Figure 6 molecules-27-02574-f006:**
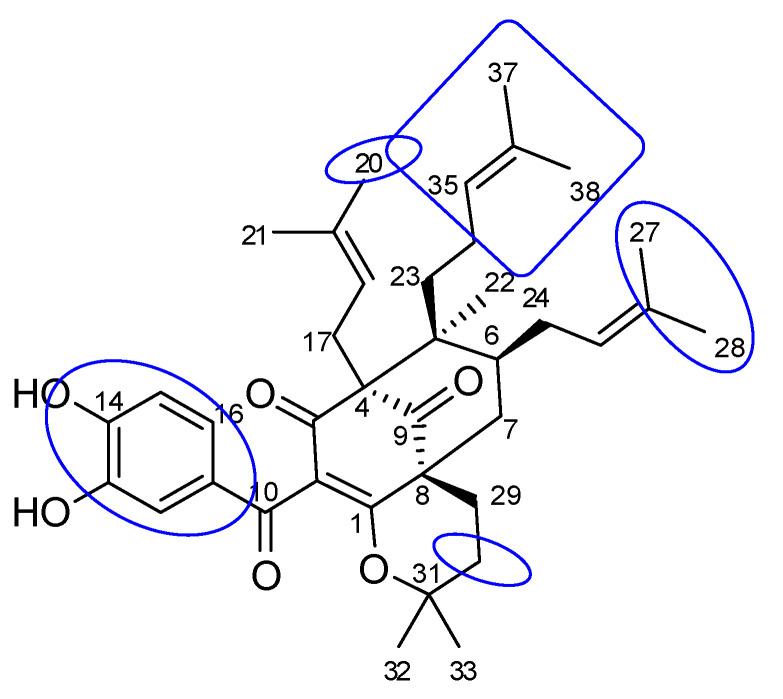
Pharmacophore of pose 148 or compound **1** showing active groups on compound **1** or pose 148.

**Figure 7 molecules-27-02574-f007:**
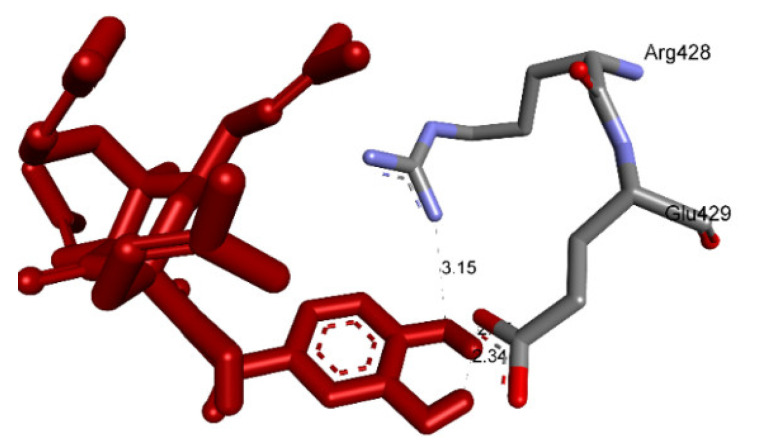
The hydrogen bonding linked from pose 41/200, compound **2**, to active atoms on enzyme 4J5T.

**Figure 8 molecules-27-02574-f008:**
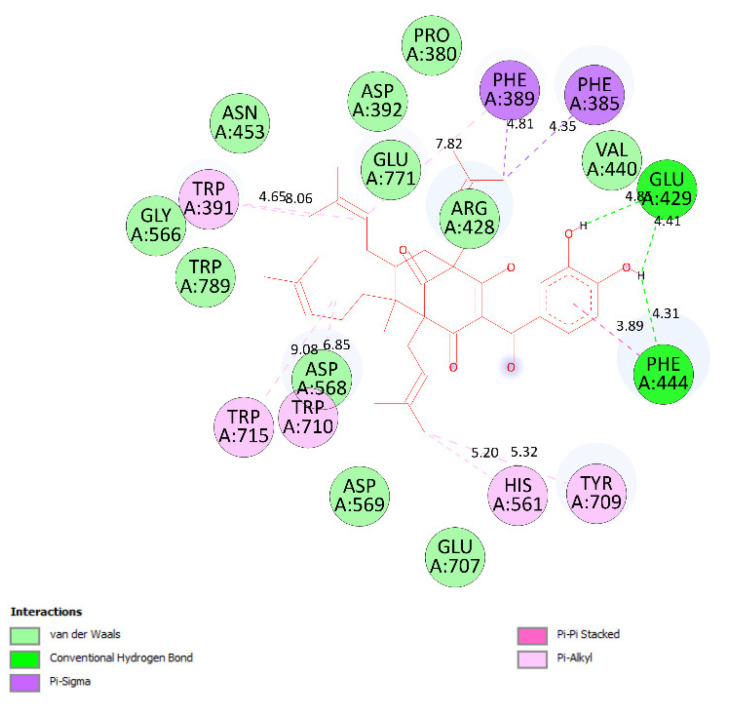
The role of ligand interactions between pose 41/200 and active residual amino acids on receptor 4J5T.

**Figure 9 molecules-27-02574-f009:**
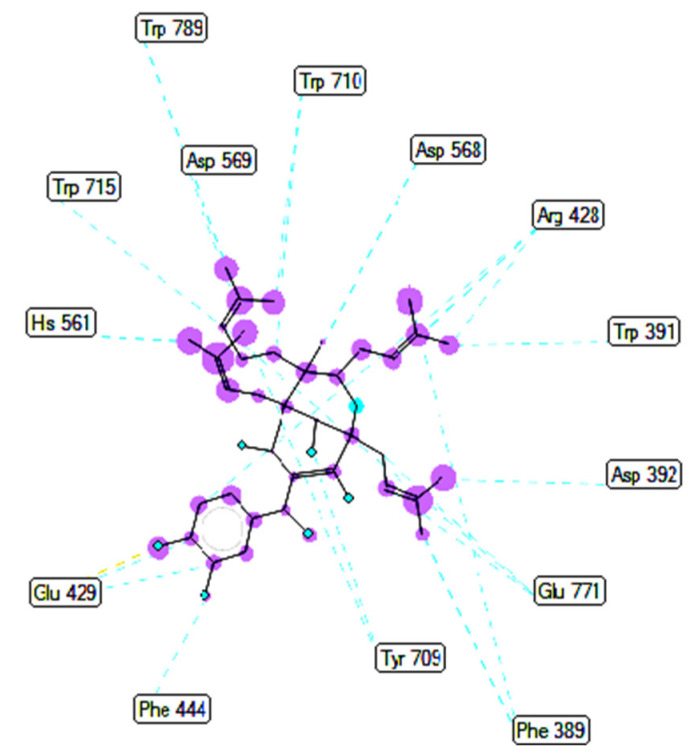
Ligand map showing the secondary interactions such as hydrogen bonding and steric and overlap interactions between residual amino acids on enzyme 4J5T and pose 41/200, compound **2**.

**Figure 10 molecules-27-02574-f010:**
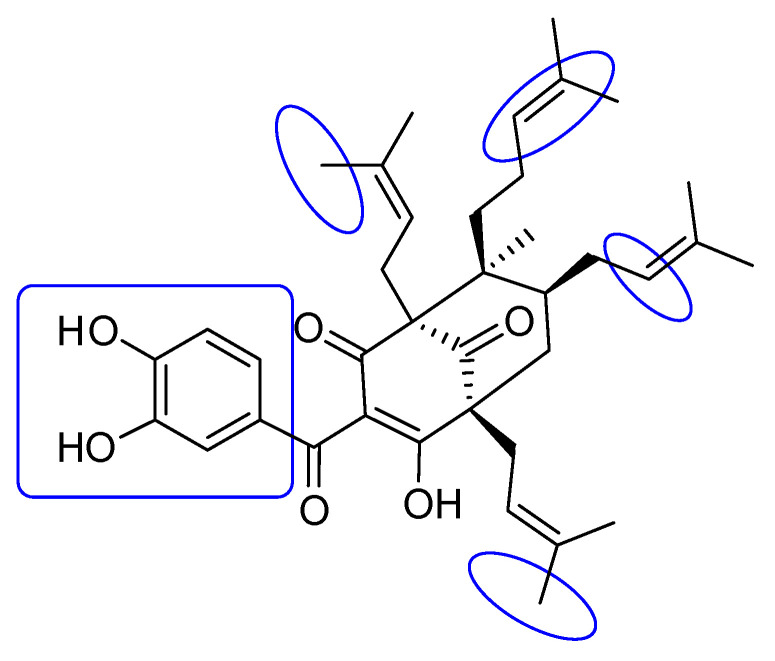
Pharmacophore of pose 41 or compound **2** indicating the active groups in compound **2** or pose 148.

**Figure 11 molecules-27-02574-f011:**
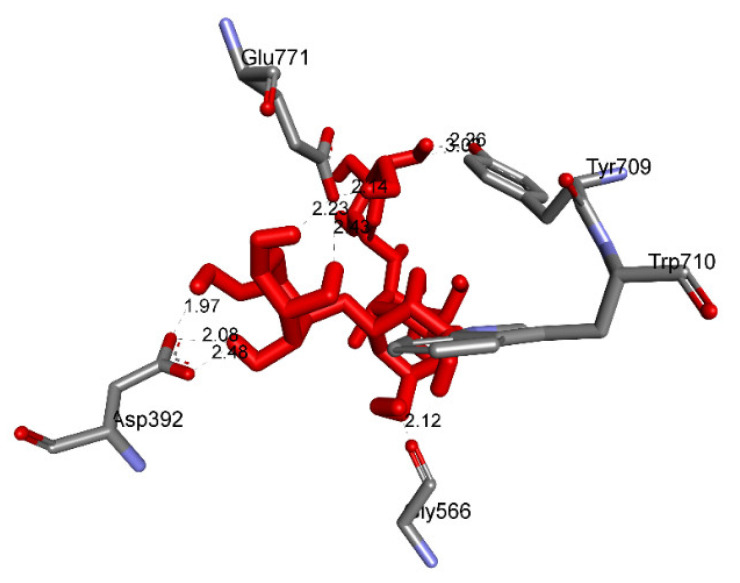
The hydrogen bonds and electrostatic interactions bound from active atoms on the ligand acarbose, a standard drug, to active atoms on residual amino acids on enzyme 4J5T.

**Figure 12 molecules-27-02574-f012:**
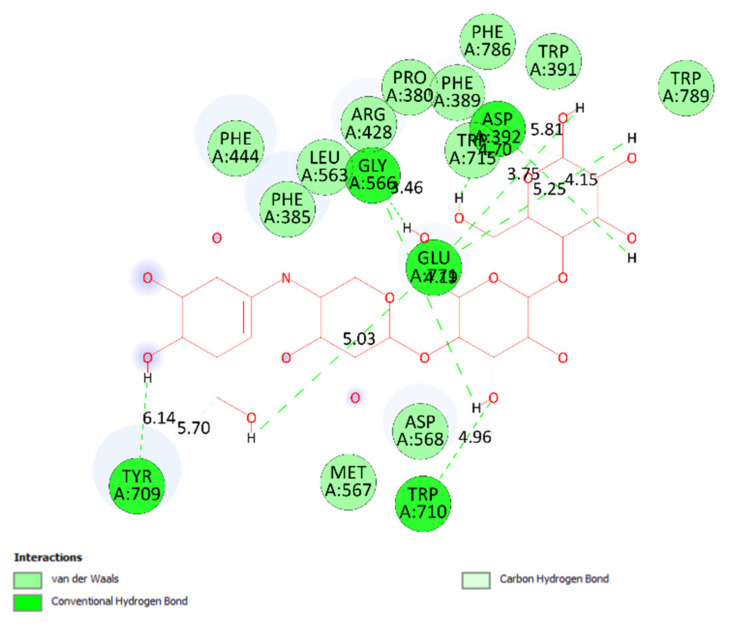
The most significant interactions formed between active atoms on pose 170/200, acarbose and active residual amino acids on enzyme 4J5T.

**Figure 13 molecules-27-02574-f013:**
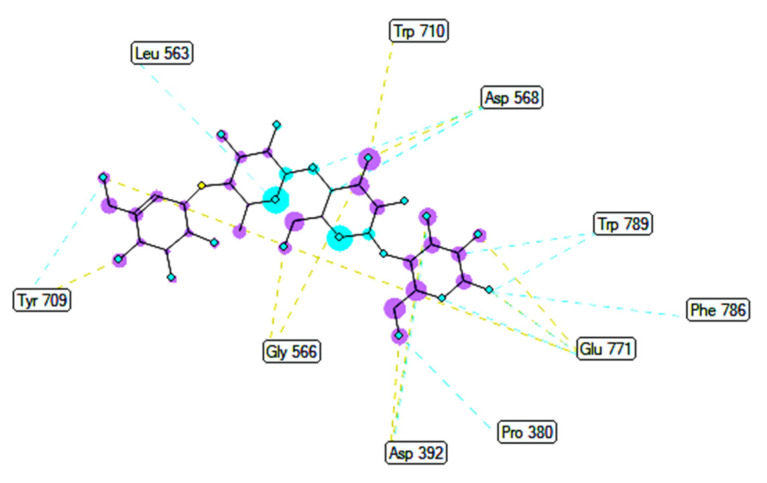
A ligand map exposing the secondary interactions such as hydrogen bonding and steric and overlap interactions between residual amino acids on enzyme 4J5T and pose 170/200, acarbose.

**Figure 14 molecules-27-02574-f014:**
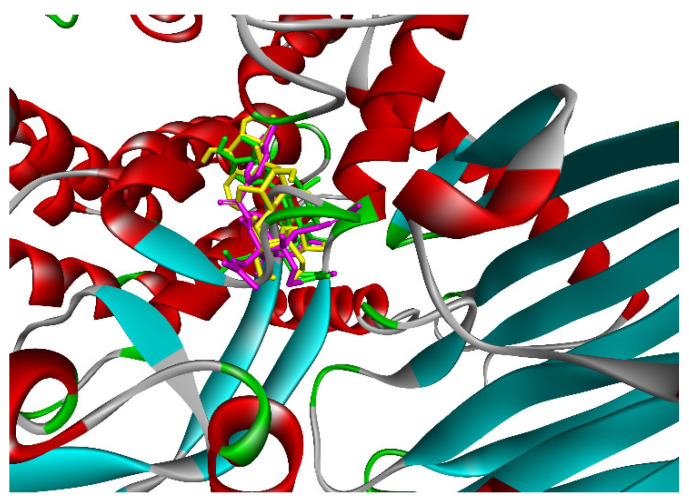
The best docking poses of ligands, pose 148 (green), pose 41 (violet), and pose 170 (yellow), docked to the same active center on enzyme 4J5T.

**Figure 15 molecules-27-02574-f015:**
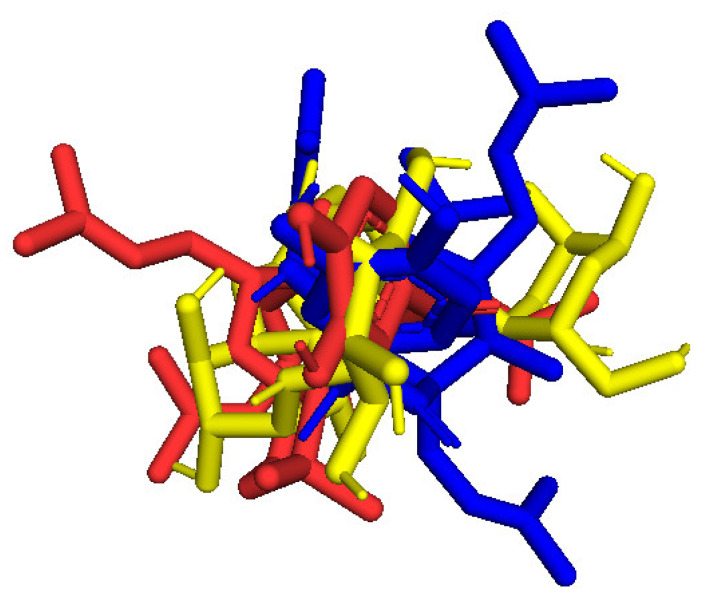
The active poses aligned to pose 170, acarbose, and RMSD calculated by PyMOL software.

**Figure 16 molecules-27-02574-f016:**
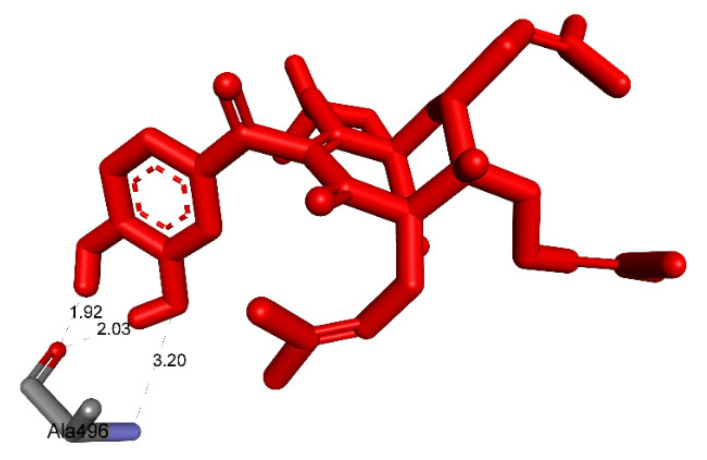
The hydrogen bonds and electrostatic interactions linked from active atoms on pose 158, compound **1**, to active residual amino acids on enzyme 2VF5.

**Figure 17 molecules-27-02574-f017:**
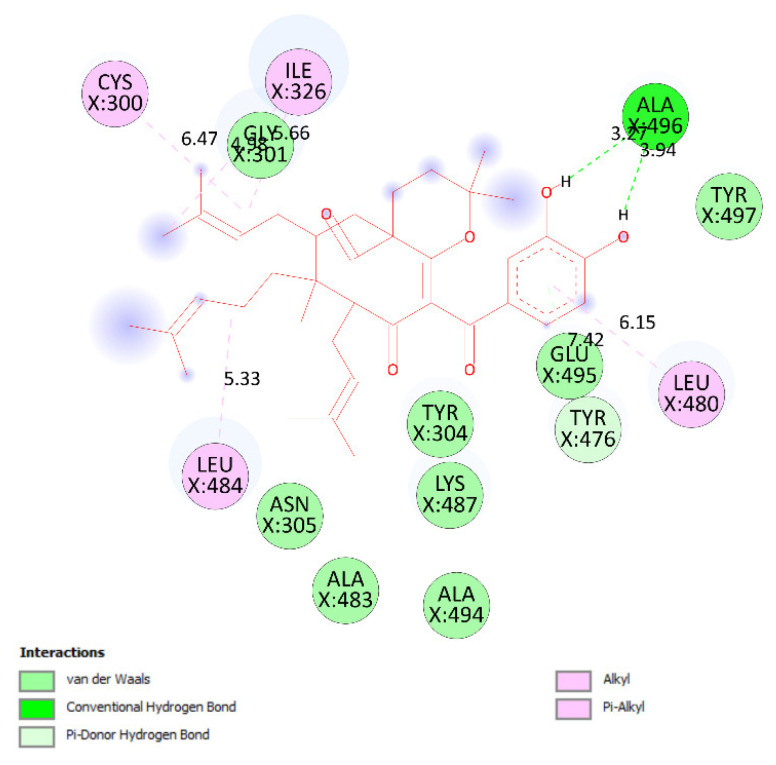
The significant ligand interactions between pose 158 and active residual amino acids on enzyme 2VF5 indicated in one 2D diagram.

**Figure 18 molecules-27-02574-f018:**
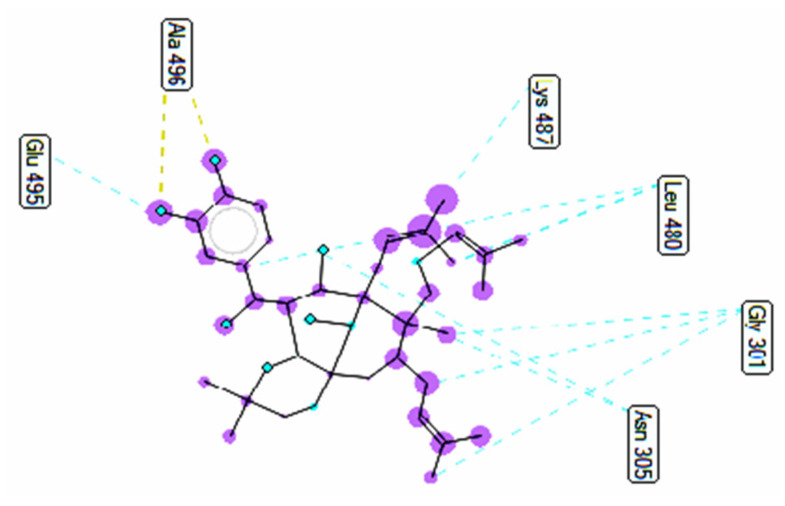
Ligand map presenting the secondary interactions between pose 158 and active residual amino acids on enzyme 2VF5.

**Figure 19 molecules-27-02574-f019:**
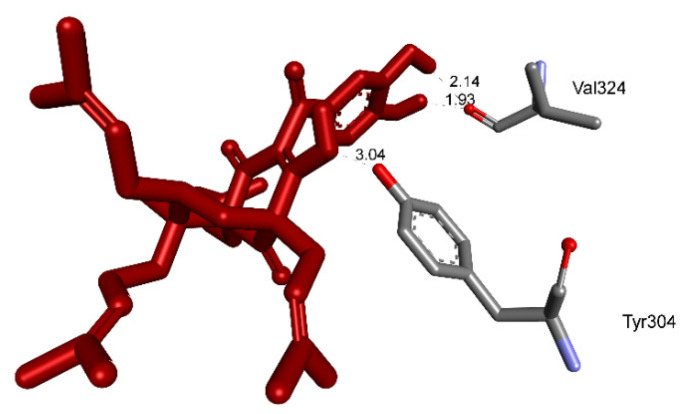
The hydrogen bonds from active atoms on pose 35/200 to active atoms on residual amino acids of enzyme 2VF5.

**Figure 20 molecules-27-02574-f020:**
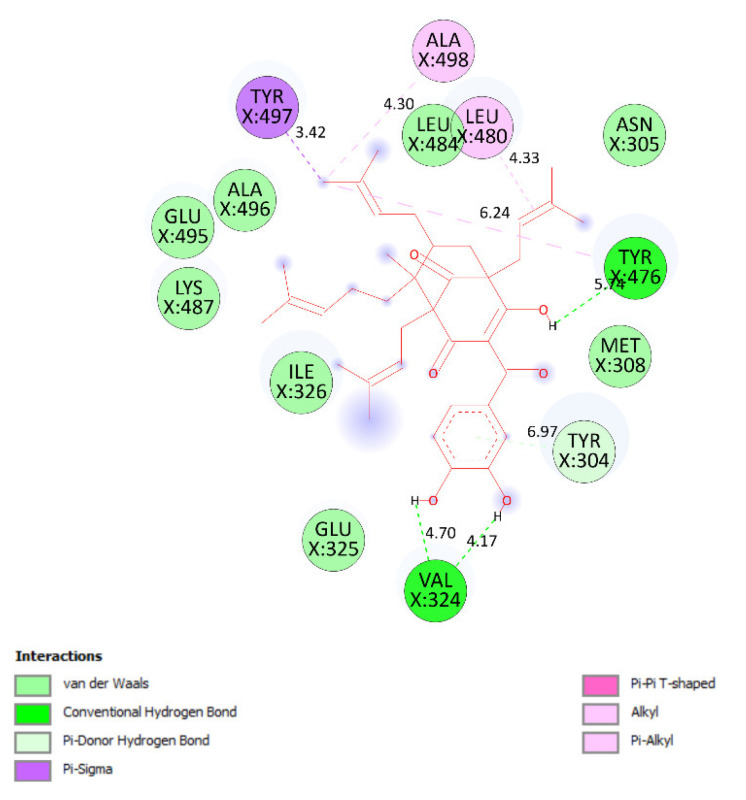
One 2D diagram exposing the significant ligand interactions between pose 35/200 and ligand 2VF5.

**Figure 21 molecules-27-02574-f021:**
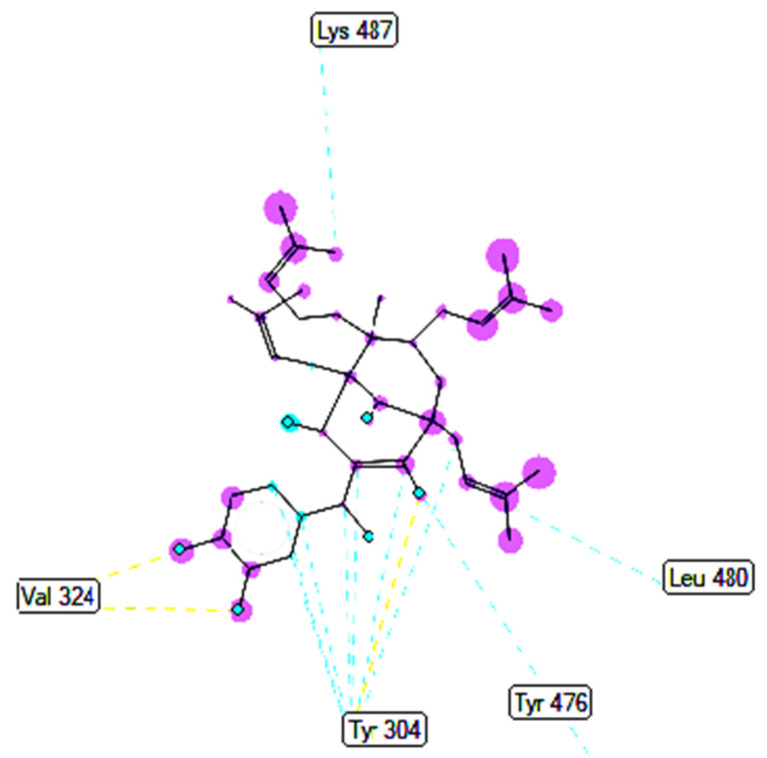
Ligand map exposing the secondary interactions between pose 35/200 and receptor 2VF5.

**Figure 22 molecules-27-02574-f022:**
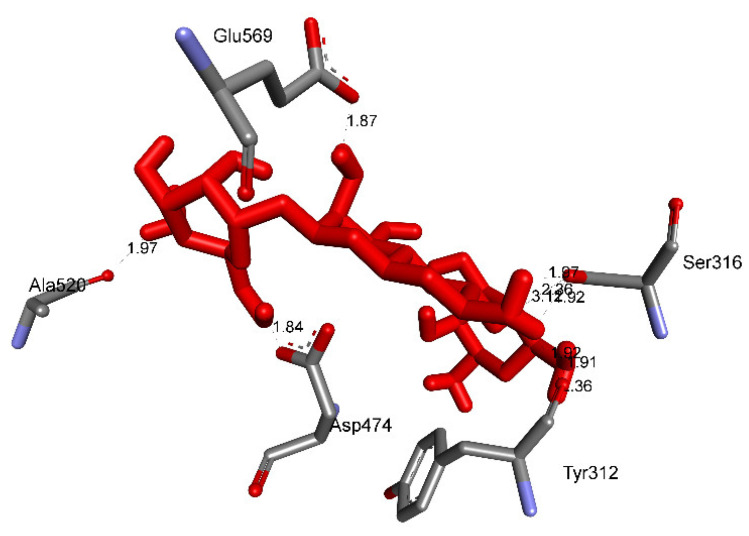
The hydrogen bonds from pose 172, apramycin, a standard drug used for antimicrobial activity, to enzyme 2VF5.

**Figure 23 molecules-27-02574-f023:**
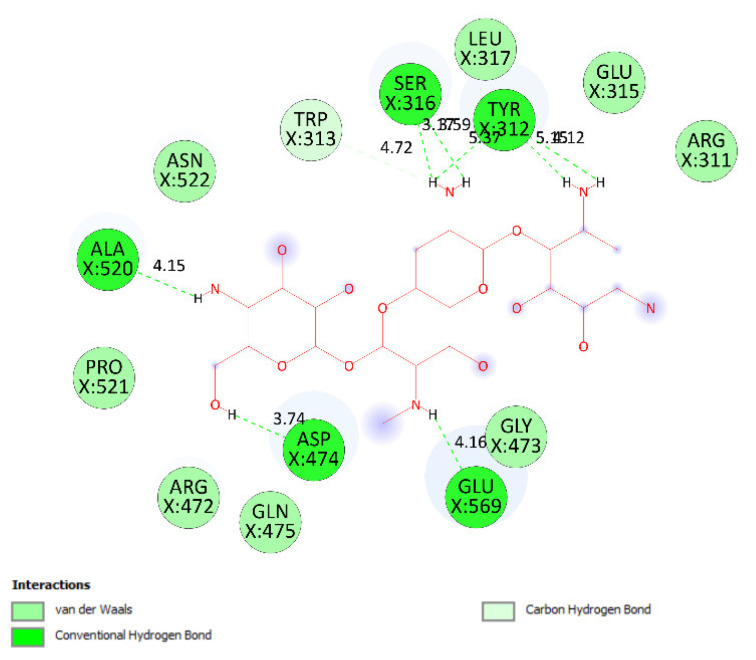
The significant ligand interactions indicated in one 2D diagram between pose 172, the standard drug apramycin, and enzyme 2VF5.

**Figure 24 molecules-27-02574-f024:**
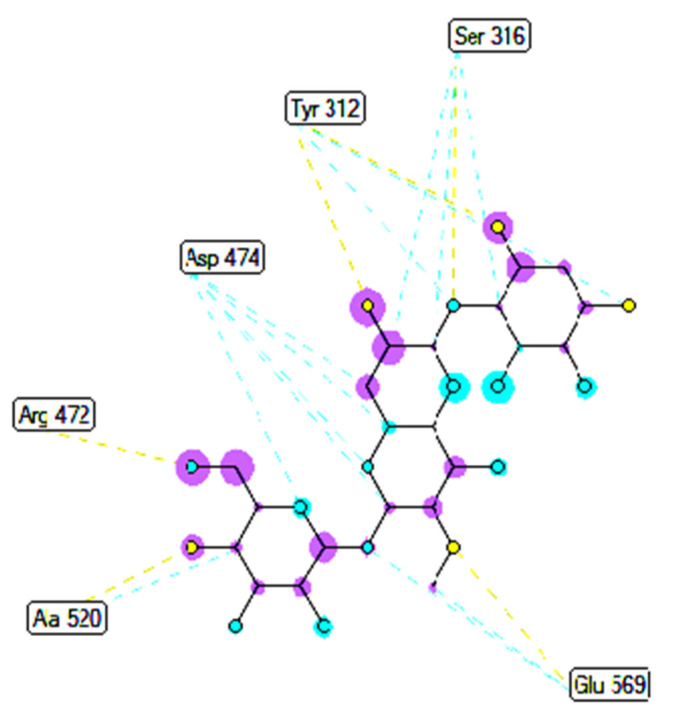
Ligand map revealing the secondary interactions formed from active atoms on pose 172, a standard drug apramycin, to residual amino acids on enzyme 2VF5.

**Figure 25 molecules-27-02574-f025:**
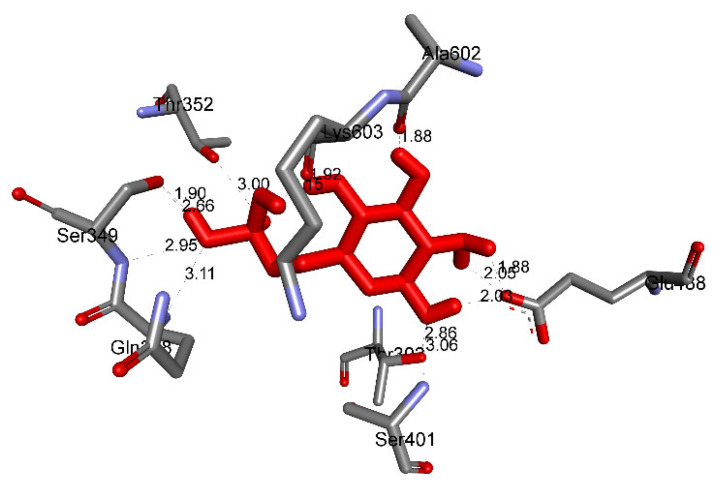
The hydrogen bonds from active atoms on pose 83, one small ligand which is available in enzyme 2VF5, to active residual amino acids on enzyme 2VF5.

**Figure 26 molecules-27-02574-f026:**
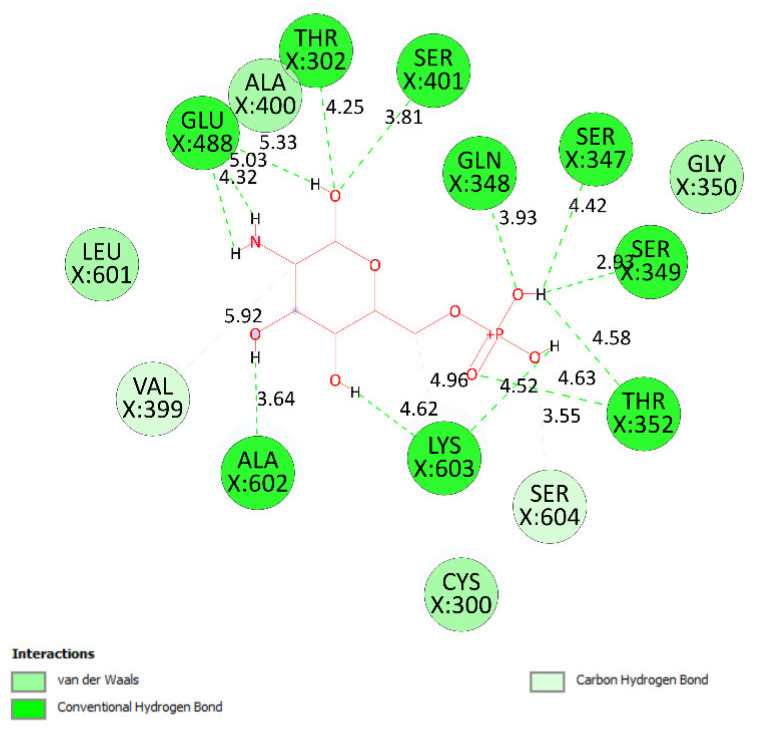
Diagram detailing the significant ligand interactions forming from pose 83, one small ligand, to active atoms on enzyme 2VF5.

**Figure 27 molecules-27-02574-f027:**
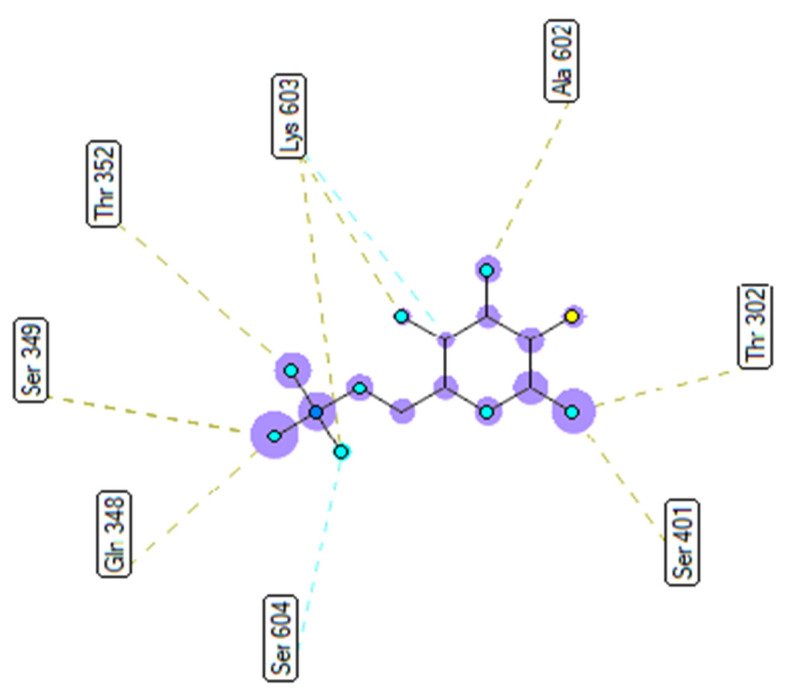
Ligand map showing the secondary interactions between pose 83, one small ligand, and enzyme 2VF5.

**Figure 28 molecules-27-02574-f028:**
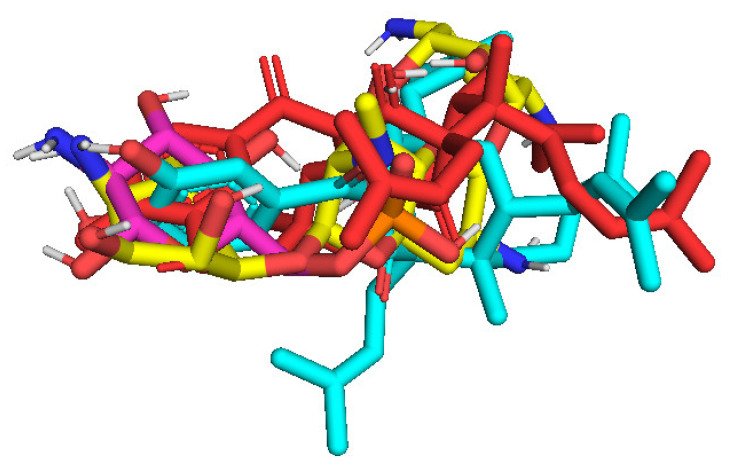
The best docking poses of the ligand, pose 158 (red), pose 35 (cyan), and pose 83 (violet) aligned to pose 172 (yellow).

**Figure 29 molecules-27-02574-f029:**
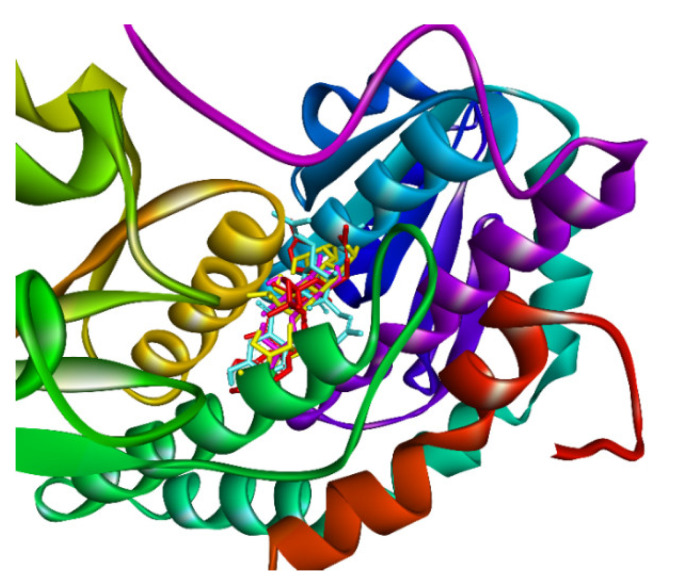
The best docking poses of ligand poses 158 (red), 35 (cyan), 83 (violet), and 172 (yellow) docked to the same active center on enzyme 2VF5.

**Table 1 molecules-27-02574-t001:** ^1^H (500 MHz) and ^13^C (125 MHz) NMR data of compounds **1** and **2** (acetone-*d*_6_, δ, ppm, J/Hz).

No	1		2	
	δ_H_	δ_C_	δ_H_	δ_C_
1		179.0		199.3
2		116.6		116.9
3		189.8		198.6
4		72.7		69.6
5		49.1		49.4
6	2.14 (m)	37.8	1.61 (m)	37.8
7	2.54 (m); 1.44 (m)	39.7	2.05 (m); 1.46 (m)	40.0
8		49.0		64.7
9		208.4		207.0
10		191.7		195.6
11		131.3		129.0
12	7.38 (d, 2.0)	116.1	7.21 (d, 2.0)	116.6
13		150.6		150.4
14		146.8		144.5
15	6.85 (d, 8.5)	115.5	6.82 (d, 8.5)	114.0
16	7.19 (dd, 8.5, 2.0)	125.6	7.21 (dd, 8.5, 2.0)	124.5
17	2.66 (dd, 13.5, 7.5); 2.50 (m)	24.7	2.81 (dd, 14.5, 9.0); 2.69 (m)	25.6
18	4.84 (brt)	122.2	4.94 (brt)	120.0
19		133.1		132.9
20	1.64 (s)	25.8	1.67 (s)	25.0
21	1.59 (s)	17.8	1.59 (s)	17.4
22	0.82 (s)	15.5	0.88 (s)	16.9
23	1.81 (m); 1.78 (m)	36.0	1.87 (m); 1.78 (m)	36.0
24	2.14 (m); 1.98 (m)	24.0	2.14 (m); 2.02 (m)	24.0
25	5.04 (brt)	123.7	5.03 (brt)	124.5
26		133.8		131.0
27	1.75 (s)	25.0	1.67 (s)	25.0
28	1.66 (s)	17.2	1.61 (s)	17.2
29	2.53 (m)	25.8	2.57 (m); 2.53 (m)	30.6
30	1.43 (m)	29.1	5.10 (brt)	120.4
31		82.0		133.8
32	1.25 (s)	31.4	1.72 (s)	25.3
33	1.01 (s)	26.7	1.69 (s)	17.3
34	2.26 (m); 1.91 (m)	31.0	2.14 (m); 2.02 (m)	31.8
35	5.21 (brt)	125.6	5.20 (brt)	122.6
36		131.7		132.9
37	1.63 (s)	25.2	1.72 (s)	25.2
38	1.59 (s)	17.3	1.67 (s)	17.3

**Table 2 molecules-27-02574-t002:** The significant calculation results for the in silico molecular docking α-glucosidase enzyme inhibition model of compounds or ligands to one receptor, α-glucosidase enzyme 4J5T (PDB).

Entry	Active Pose	Affinity Energy (a)	*K*_i_ (b)	The Number of Hydrogen Bonds (c)	The Property and Bond Length (d)
Compound **1**	148/200	−10.51	0.02	2	A:Arg428:N−Compound **1**:O (3.01 Å) Compound **1**:H−A:Glu429:O (2.39Å)
Compound **2**	41/200	−10.12	0.04	3	A:Arg428:N−Compound **2**:O (3.16 Å) Compound **2**:H−A:Glu 429:O (2.15 Å) Compound **2**:H−A:Glu 429:O (2.34 Å)
Acarbose	170/200	−5.22	149.6	10	A: Tyr709:O−Acarbose:O (3.08 Å) A:Trp710:N−Acarbose:O (2.69 Å) Acarbose:H−A:Tyr709:OH (2.26 Å) Acarbose:H−A:Glu771:O (2.14 Å) Acarbose:H−A:Gly566:O (2.12 Å) Acarbose:H−A:Asp 392:O (1.97 Å) Acarbose:H−A:Glu771:O (2.23 Å) Acarbose:H−A:Glu771:O (2.43 Å) Acarbose:H−A:Asp392:O (2.48 Å) Acarbose 1:H−A:Asp392:O(2.08 Å)

(a) In units of kcal·mol^−1^ from the Auto Dock Tools (ADT) package. (b) Inhibition constant in units of µM and calculated by ADT. (c) From the Discovery Studio (DSC) package after completion of calculated docking. (d) Calculated by the ADT package and visualized by the DSC package in angstroms.

**Table 3 molecules-27-02574-t003:** The values of RMSD between pose pairs, with acarbose as a reference pose.

RMSD (Å)	Pose 148	Pose 41	Pose 170
Pose 170, a reference pose	2.28	4.094	0

**Table 4 molecules-27-02574-t004:** The essential calculation results for the in silico docking model of ligands such as compounds **1** and **2**, apramycin, and small ligand in receptor 2VF5 to the receptor 2VF5 (PDB).

Entry	Active Pose	Affinity Energy (a)	*K*_i_ (b)	The Number of Hydrogen Bonds (c)	The Property and Bond Length (d)
Compound **1**	158	−8.56	0.53	3	X:Ala 496:N−Compound **1**:O (3.2 Å) Compound **1**:H−X:Ala496:O (1.92 Å) Compound **1**:H−X:Ala 496:O (2.03 Å)
Compound **2**	35	−6.24	26.84		X:Tyr304:OH−Compound **2**:O (3.04 Å) Compound **2**:H−X:Val 324:O (1.93 Å) Compound **2**:H−X:Val 324:O (2.14 Å)
Apramycin	172	−6.94	8.17	10	X:Ser316:O−Apramycin:N (2.36 Å) X:Ser316:O−Apramycin:O (3.12 Å) Apramycin:H−X:Ala 520:O (1.96 Å) Apramycin:H−X:Asp474:OD1 (1.84 Å) Apramycin:H−X:Glu569:OE2 (1.87 Å) Apramycin:H−X:Tyr312:O (1.91 Å) Apramycin:H−X:Ser316:OG (1.92 Å) Apramycin:H −X:Ser316:OG (1.97 Å) Apramycin:H−X:Tyr312:O (1.91 Å) Apramycin:H−X:Tyr312:O (2.36 Å)
Small ligand	83	−5.38	114	13	X:Thr 302:O−Small ligand:O (2.86 Å) X:Gln 348:N−Small ligand:O (3.11 Å) X:Ser 349:N−Small ligand:O (2.95 Å) X:Ser 349:O−Small ligand:O (2.66 Å) X:Thr 352:O−Small ligand:O (3.00 Å) X:Ser 401:N−Small ligand:O (3.06 Å) Small ligand:H−X:Glu 488:O (2.03 Å) Small ligand:H−X:Glu 488:O (1.88 Å) Small ligand:H−X:Glu 488:O (2.05 Å) Small ligand:H−X:Ala 602:O (1.88 Å) Small ligand:H−X:Ser 349:O (1.90 Å) Small ligand:H−X:Lys 603:O (2.15 Å) Small ligand:H−X:Lys 603:O (1.92 Å)

(a) In units of kcal·mol^−1^ from the Auto Dock Tools (ADT) package. (b) Inhibition constant in units of µM and calculated by ADT. (c) From the Discovery Studio (DSC) package after completion of calculated docking. (d) Calculated by the ADT package and visualized by the DSC package in angstroms.

**Table 5 molecules-27-02574-t005:** The values of RMSD between pairs of poses, with apramycin, a standard drug, as a reference pose.

MSD (Å)	Pose 158	Pose 35	Pose 83	Pose 172
Pose 172, a standard drug, a reference pose	3.32	2.166	2.839	0

**Table 6 molecules-27-02574-t006:** Physicochemical properties of compound **1**.

Property	Value	Comment
Molecular weight	602.36	Contain hydrogen atoms. Optimal: 100–600
Volume	657.955	Van der Waals volume
nHA	6	Number of hydrogen bond acceptors. Optimal: 0–12
nHD	2	Number of hydrogen bond donors. Optimal: 0–7
nRot	9	Number of rotatable bonds. Optimal: 0–11
MaxRing	12	Number of atoms in the biggest ring. Optimal: 0–18
nHet	6	Number of heteroatoms. Optimal: 1–15
fChar	0	Formal charge. Optimal: −4–4
nRig	27	Number of rigid bonds. Optimal: 0–30
Flexibility	0.333	Flexibility = nRot/nRig
Stereocenters	4	Optimal: ≤2
Optimal: ≤2	100.9	Topological polar surface area. Optimal: 0–140
logS	−4.218	Log of the aqueous solubility. Optimal: −4–0.5 log mol/L
logP	8.424	Log of the octanol/water partition coefficient. Optimal: 0–3
logD	5.451	logP at physiological pH 7.4. Optimal: 1–3

**Table 7 molecules-27-02574-t007:** Medicinal chemistry of compound **1**.

Property	Value	Comment
QED	0.096	A measure of drug-likeness based on the concept of desirability; attractive: >0.67; unattractive: 0.49~0.67; too complex: <0.34.
SAscore	6.097	Synthetic accessibility score is designed to estimate ease of synthesis of drug-like molecules. SAscore ≥ 6, difficult to synthesize; SAscore < 6, easy to synthesize.
Fsp^3^	0.553	The number of sp^3^ hybridized carbons/total carbon count, correlating with melting point and solubility. Fsp^3^ ≥ 0.42 is considered a suitable value.
MCE-18	153.763	MCE-18 stands for medicinal chemistry evolution. MCE-18 ≥ 45 is considered a suitable value.
NPscore	2.295	Natural-product-likeness score. This score is typically in the range of −5 to 5. The higher the score is, the higher the probability is that the molecule is an NP.
Lipinski Rule	Rejected	MW ≤ 500; logP ≤ 5; Hacc ≤ 10; Hdon ≤ 5. If two properties are out of range, a poor absorption or permeability is possible; one property being out of range is acceptable.
Pfizer Rule	Accepted	logP > 3; TPSA < 75; compounds with a high log P (>3) and low TPSA (<75) are likely to be toxic.
GSK Rule	Rejected	MW ≤ 400; logP ≤ 4; compounds satisfying the GSK rule may have a more favorable ADMET profile.
Golden Triangle	Rejected	200 ≤ MW ≤ 500; −2 ≤logD ≤ 5; compounds satisfying the Golden Triangle rule may have a more favorable ADMET profile.
PAINS	1 alert	Pan-assay interference compounds, frequent hitters, *α*-screen artifacts and reactive compound.
ALARM NMR	4 alerts	Thiol reactive compounds.
BMS	0 alerts	Undesirable, reactive compounds.
Chelator Rule	2 alerts	Chelating compounds.

**Table 8 molecules-27-02574-t008:** The absorption of compound **1**.

Property	Value	Comment
Caco-2 Permeability	−4.852	Optimal: higher than −5.15 log unit
MDCK Permeability	1.6e−05	Low permeability: <2 × 10^−^^6^ cm/s Medium permeability: 2–20 × 10^−^^6^ cm/s High passive permeability: >20 × 10^−^^6^ cm/s
Pgp-inhibitor	0.175	Category 1: inhibitor; Category 0: non-inhibitor; the output value is the probability of being Pgp-inhibitor
Pgp-substrate	0.021	Category 1: substrate; Category 0: non-substrate; the output value is the probability of being Pgp-substrate
HIA	0.038	Human intestinal absorption; Category 1: HIA+(HIA < 30%); Category 0: HIA-(HIA < 30%); the output value is the probability of being HIA+

**Table 9 molecules-27-02574-t009:** The properties of the drug distribution of compound **1**.

Property	Value	Comment
PPB	95.95%	Plasma protein binding; optimal: <90%. Drugs with high protein binding may have a low therapeutic index.
VD	0.994	Volume distribution; optimal: 0.04–20 L/kg.
BBB Penetration	0.012	Blood–brain barrier penetration; Category 1: BBB+; Category 0: BBB−; the output value is the probability of being BBB+.
Fu	7.588%	The fraction unbound in plasma; low: <5%; middle: 5~20%; high: >20%.

**Table 10 molecules-27-02574-t010:** The properties of the drug metabolism of compound **1**.

Property	Value	Comment
CYP1A2 inhibitor	0.1	Category 1: inhibitor; Category 0: non-inhibitor; the output value is the probability of being an inhibitor.
CYP1A2 substrate	0.26	Category 1: substrate; Category 0: non-substrate; the output value is the probability of being a substrate
CYP2C19 inhibitor	0.802	Category 1: inhibitor; Category 0: non-inhibitor; the output value is the probability of being an inhibitor
CYP2C19 substrate	0.352	Category 1: substrate; Category 0: non-substrate; the output value is the probability of being a substrate
CYP2C9 inhibitor	0.762	Category 1: inhibitor; Category 0: non-inhibitor; the output value is the probability of being an inhibitor.
CYP2C9 substrate	0.882	Category 1: substrate; Category 0: non-substrate; the output value is the probability of being a substrate.
CYP2D6 inhibitor	0.905	Category 1: inhibitor; Category 0: non-inhibitor; the output value is the probability of being an inhibitor.
CYP2D6 substrate	0.022	Category 1: substrate; Category 0: non-substrate; the output value is the probability of being a substrate.
CYP3A4 inhibitor	0.866	Category 1: inhibitor; Category 0: non-inhibitor; the output value is the probability of being an inhibitor.
CYP3A4 substrate	0.854	Category 1: substrate; Category 0: non-substrate; the output value is the probability of being a substrate.

**Table 11 molecules-27-02574-t011:** The properties of the drug excretion of compound **1**.

Property	Value	Comment
CL	20.163	Clearance; high: >15 mL/min/kg; moderate: 5–15 mL/min/kg; low: <5 mL/min/kg.
T1/2	0.027	Category 1: long half-life; Category 0: short half-life; long half-life: >3 h; short half-life: <3 h; the output value is the probability of having a long half-life.

**Table 12 molecules-27-02574-t012:** The properties of the drug toxicity of compound **1**.

Property	Value	Comment
hERG Blockers	0.004	Category 1: active; Category 0: inactive; the output value is the probability of being active.
H-HT	0.956	Human hepatotoxicity; Category 1: H-HT positive (+); Category 0: H-HT negative (−); the output value is the probability of being toxic.
DILI	0.853	Drug-induced liver injury. Category 1: drugs with a high risk of DILI; Category 0: drugs with no risk of DILI. The output value is the probability of being toxic.
AMES Toxicity	0.014	Category 1: AMES positive (+); Category 0: AMES negative (−); the output value is the probability of being toxic.
Rat Oral Acute Toxicity	0.432	Category 0: low toxicity; Category 1: high toxicity; the output value is the probability of being highly toxic.
FDAMDD	0.037	Maximum recommended daily dose; Category 1: FDAMDD (+); Category 0: FDAMDD (−); the output value is the probability of being positive.
Skin Sensitization	0.013	Category 1: sensitizer; Category 0: non-sensitizer; the output value is the probability of being a sensitizer.
Carcinogencity	0.539	Category 1: carcinogens; Category 0: non-carcinogens; the output value is the probability of being toxic.
Eye corrosion	0.003	Category 1: corrosive; Category 0: noncorrosive; the output value is the probability of being corrosive.
Eye irritation	0.025	Category 1: irritants; Category 0: non-irritants; the output value is the probability of being an irritant.
Respiratory Toxicity	0.942	Category 1: respiratory toxicants; Category 0: respiratory non-toxicants; the output value is the probability of being toxic.

**Table 13 molecules-27-02574-t013:** The properties of the environmental toxicity of compound **1**.

Property	Value	Comment
Bioconcentration Factors	0.444	Bioconcentration factors are used for considering secondary poisoning potential and assessing risks to human health via the food chain. The unit is −log10((mg/L)/(1000 × MW)).
IGC_50_	4.596	Tetrahymena pyriformis 50 percent growth inhibition concentration; the unit is −log10((mg/L)/(1000 × MW)).
LC_50_FM	6.123	96-h fathead minnow 50 percent lethal concentration; the unit is −log10((mg/L)/(1000 × MW)).
LC^50^DM	6.105	48-h daphnia magna 50 percent lethal concentration; the unit is −log10((mg/L)/(1000 × MW)).

**Table 14 molecules-27-02574-t014:** Toxicophore rules of compound **1**.

Property	Value	Comment
Acute Toxicity Rule	0 alerts	20 substructures; acute toxicity during oral administration
Genotoxic Carcinogenicity Rule	1 alert	117 substructures; carcinogenicity or mutagenicity
Nongenotoxic Carcinogenicity Rule	0 alerts	23 substructures; carcinogenicity through nongenotoxic mechanisms
Skin Sensitization Rule	12 alerts	155 substructures; skin irritation
Aquatic Toxicity Rule	3 alerts	99 substructures; toxicity to liquid (water)
Nonbiodegradable Rule	3 alerts	19 substructures; nonbiodegradable
SureChEMBL Rule	0 alerts	164 substructures; MedChem unfriendly status

## Data Availability

All data supporting this study are available in the manuscript and the Appendix A.

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
