# Peer review of "α-Glucosidase Inhibitory and Antimicrobial Benzoylphloroglucinols from Garcinia schomburgakiana Fruits: In Vitro and In Silico Studies"

_molecules, 2022, doi:10.3390/molecules27082574_

Round 1
Reviewer 1 Report
This manuscript describes new polyprenylated benzoylphloroglucinol, isolated from the fruits of Garcinia schomburgkiana as potential antidiabetic compound. In silico and in vitro testing were performed to analyze α-glucosidase inhibition as well as antibacterial action. The manuscript is robust and fit within the scopes of the journal. Therefore, this referee believes that it deserves to be published in Molecules, pending the following minor points:
-Authors should comment in the proper section, the common side effects provoked by alpha-glucosidase inhibitors
- In the introduction section, the description of antibacterial agents and docking methods are out of place and do not have adequate coherence. Please rewrite this section.
-Authors should report the MIC values using micromolar concentrations [mM] instead of micrograms per mL [mg/mL], in order to compare relative potencies with positive control Kanamycin
-Docking Score is proper than Free Gibbs energy to report the Affinity Energy values
-The correct abbreviation of “AutoDock Tools” is “ADT” (please remove ATD)
-The quality of figures 4,5,7,8,9,12 and 13 is poor. The resolution must be increased
-The in silico calculations must be completed with the pharmacokinetic and toxicological predictions of 1, using a free software such as ADMETLab2.0 or Pre ADMET
-The English of the manuscript must be improved
Author Response
Thank you very much for your valuable and positive comments. They are very helpful in proving the quality of our manuscript.
We have revised the expression according to your comments. The revised parts are highlighted in yellow for your convenience of re-reviewing.
|
Reviewers' Comments |
Response to reviewers' Comments |
|
Reviewer 1 |
|
|
This manuscript describes new polyprenylated benzoylphloroglucinol, isolated from the fruits of Garcinia schomburgkiana as potential antidiabetic compound. In silico and in vitro testing were performed to analyze α-glucosidase inhibition as well as antibacterial action. The manuscript is robust and fit within the scopes of the journal. Therefore, this referee believes that it deserves to be published in Molecules, pending the following minor points: -Authors should comment in the proper section, the common side effects provoked by alpha-glucosidase inhibitors - In the introduction section, the description of antibacterial agents and docking methods are out of place and do not have adequate coherence. Please rewrite this section. -Authors should report the MIC values using micromolar concentrations [mM] instead of micrograms per mL [mg/mL], in order to compare relative potencies with positive control Kanamycin -Docking Score is proper than Free Gibbs energy to report the Affinity Energy values -The correct abbreviation of “AutoDock Tools” is “ADT” (please remove ATD) -The quality of figures 4,5,7,8,9,12 and 13 is poor. The resolution must be increased -The in silico calculations must be completed with the pharmacokinetic and toxicological predictions of 1, using a free software such as ADMETLab2.0 or Pre ADMET -The English of the manuscript must be improved
|
We sincerely thank the reviewer for pointing out these mistakes in our manuscript, and they are very helpful in proving the quality of our manuscript. We have corrected the errors accordingly.
We have revised and adjusted the text where highlighted.
We have added the MIC value of antimicrobial activity and the IC50 value of cytotoxicity against the HEK293 normal cell line.
This revised manuscript has been read and checked for English usage and grammar by a native English editor (johnsidneywinward@gmail.com) from the English Clinic, Faculty of Science and Technology, Thammasat University.
|
Reviewer 2 Report
The authors describe herein the identification of a new polyprenylated benzoylphloroglucinol compound extracted from the fruits of G. schomburgakiana and its evaluation as an alpha-glucosidase inhibitor and antibacterial agent. Docking studies were also conducted to investigate its mechanism of interaction.
I have some remarks and suggestions to bring to the attention of the authors:
- The emphasis on long-time period diabetic headaches in the first sentence is quite unexpected, diabetes being a metabolic disease characterized by otherwise more prevalent disorders and by a risk of serious complications or "metabolic accidents" (as stated below).
- line 42-43: please write « alpha-glucosidases perform... » in plural. The same goes for line 44.
- line 48: acarbose could also be cited as an alpha-glucosidase inhibitor in recent use.
- line 61-62: the last sentence of this paragraph is also a bit unexpected. It would benefit from being better articulated with the previous sentence to make the point more coherent.
- line 63-65: same remark as before, the speech lacks coherence as two statements with apparently no connection between them are successively stated without transition.
- line 75 and line 77: bibliographical references are expected at the end of these two sentences to support them.
- line 88: please replace « as weel as the » by « and ».
- line 92: has the mechanism of inhibition been formally elucidated following this study? If not, please replace "elucidate" by "investigate".
- line 140: please write « Biological activites »
- line 145-146: same remark as before, the last sentence of this paragraph is quite unexpected. A clearer rephrasing, saying that the lack of literature data on this point does not allow to relate to the results of other similar compounds, would be welcome.
- Paragraph 2.2.: concerning the antimicrobial activity of 1 and 2, what about their toxicity on healthy cells? It is difficult to impute antibacterial properties to a molecule without having verified its absence of cytotoxic effect beforehand, otherwise many toxic compounds would be considered excellent antibiotics.
- line 147: please reword this sentence to be gramatically correct
- line 149 and 150: please remove « the » before « inhibition zones »
- line 174: please add « are » before « depicted »
- line 176: please replace the dot after ref. 31 by a coma
- line 176: please write « capping unit » without capital letter
- Paragraphs 2.4.1 and 2.4.2 are extremely unclear and difficult to understand. A methodical revision of these paragraphs, both on the form of the speech and on its organization, is necessary to facilitate understanding.
- The multiple Figures proposed are informative, but also spoilt by their large number. Wouldn't it be possible to reformat them into figures divided into sub-images, with each individual part labeled by a letter?
- line 433: please replace the dot by a coma
- line 498: please write « and » in normal font and not in italics
- line 500: please write « in silico » without capital letter
Author Response
|
We sincerely thank the reviewer for pointing out these mistakes in our manuscript, and they are very helpful in proving the quality of our manuscript. We have corrected the errors accordingly.
We have revised and adjusted the text where highlighted.
We have added the MIC value of antimicrobial activity and the IC50 value of cytotoxicity against the HEK293 normal cell line.
This revised manuscript has been read and checked for English usage and grammar by a native English editor (johnsidneywinward@gmail.com) from the English Clinic, Faculty of Science and Technology, Thammasat University.
|

Round 2
Reviewer 2 Report
I aknowledge the changes made by the authors and thank them for the modifications made on the manuscript.